# Reconsidering Spatial Alignment for Longitudinal Breast Cancer Risk Prediction

Solveig Thrun[*1], Stine Hansen[2], Zijun Sun[3], Nele Blum[4], Suaiba A. Salahuddin[1], Xin Wang[5], Kristoffer Wickstrøm[1], Elisabeth Wetzer[1], Robert Jenssen[1,6,7], Maik Stille[4], and Michael Kampffmeyer[1,6]

[1]Department of Physics and Technology, UiT The Arctic University of Norway, Tromsø, Norway
[2]SPKI The Norwegian Centre for Clinical Artificial Intelligence, University Hospital of North Norway, Tromsø, Norway
[3]Department of Computer Science and Engineering, University of Bologna, Bologna, Italy
[4]Fraunhofer Research Institution for Individualized and Cell-Based Medical Engineering IMTE, Lübeck, Germany
[5]Department of Radiology, Netherlands Cancer Institute (NKI), Amsterdam, The Netherlands
[6]Norwegian Computing Center, Oslo, Norway
[7]Pioneer Centre for AI, University of Copenhagen, Copenhagen, Denmark

## Abstract

Regular mammography screening is vital for early breast cancer detection, and deep learning enables more personalized strategies. However, misalignment across time points can obscure subtle tissue changes and reduce prediction accuracy. This study evaluates image-based, feature-level, and implicit alignment methods on two large mammography datasets, showing that our proposed image-based registration model achieves the highest accuracy and anatomically plausible deformations, highlighting the importance of precise alignment in longitudinal risk prediction.

## 1 Introduction

Mammography remains the gold standard for breast cancer screening [1], effectively reducing mortality [2]. Recent deep learning studies show that incorporating longitudinal mammography—images from multiple timepoints—can enhance risk prediction beyond single-timepoint models [3–7]. Realizing these benefits requires accurate alignment across time, complicated by variations in breast tissue and differences in patient positioning [8]. Alignment strategies are typically categorized as either explicit, where images or features are directly registered, or implicit, where alignment is learned jointly during feature extraction. We perform the first systematic study of alignment strategies for longitudinal breast cancer risk prediction, providing insights into both explicit and implicit approaches. Building on these insights, we propose a new image-based alignment model that achieves improved predictive performance. **Our main contributions are:**

- A unified framework for evaluating explicit (image-/feature-level) and implicit alignment strategies for longitudinal breast cancer risk

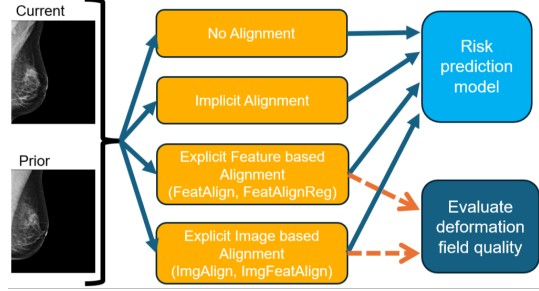

**Figure 1.** Overview of the proposed longitudinal risk prediction framework, highlighting the alignment strategies assessed within a unified risk prediction approach.

prediction.
- A novel risk prediction model that leverages image-based alignment to generate anatomically meaningful deformations, achieving state-of-the-art performance on two large-scale datasets.

## 2 Methods

We address the challenge of five-year breast cancer risk prediction by evaluating six temporal alignment strategies within a unified risk prediction framework (Figure 1).

**No Alignment:** Our baseline builds on prior work [7, 9], combining Multilevel Joint Learning [7], Temporal Self-Attention [10], and a Cumulative Probability Layer [4, 9, 11]. Current and prior images are encoded with a shared backbone, processed via temporal self-attention, and used for risk prediction. Additional prediction heads estimate risk from each timepoint independently.

**Implicit Alignment:** In this strategy, current and prior images are encoded, and their feature maps are concatenated before being processed through convolutional and attention layers. Temporal dependencies are learned implicitly, without explicit spatial alignment.

---

*Corresponding Author.

**Table 1.** 1–5 year breast cancer risk prediction using different alignment methods. C-index and selected AUC values (1, 3, 5 years) with 95% confidence intervals for both datasets.

| Method | EMBED | | | | CSAW-CC | | | |
|---|---|---|---|---|---|---|---|---|
| | C-index ↑ | 1-yr ↑ | 3-yr ↑ | 5-yr ↑ | C-index ↑ | 1-yr ↑ | 3-yr ↑ | 5-yr ↑ |
| NoAlign | 64.0 (61.7–66.7) | 64.9 (62.1–67.9) | 63.7 (61.2–66.3) | 55.7 (51.4–60.0) | 65.9 (64.0–67.8) | 66.1 (63.8–68.3) | 65.7 (63.8–67.6) | 66.8 (64.5–68.9) |
| Implicit | 70.9 (68.6–73.3) | 72.5 (69.3–75.5) | 69.3 (66.6–71.8) | 65.7 (62.0–69.7) | 67.6 (65.8–69.7) | 68.2 (65.7–70.6) | 68.3 (66.3–70.2) | 68.7 (66.3–71.1) |
| FeatAlign | 72.2 (69.5–75.5) | 72.4 (69.5–75.6) | 72.0 (69.7–74.6) | 68.5 (64.8–72.0) | 69.1 (67.0–71.1) | 70.1 (67.9–72.4) | 70.0 (68.1–71.9) | 71.6 (69.4–73.8) |
| FeatAlignReg | 70.6 (67.8–73.2) | 71.2 (68.3–74.3) | 70.7 (68.2–73.5) | 65.7 (61.7–69.6) | 68.4 (66.4–70.4) | 68.9 (66.7–71.2) | 69.8 (68.0–71.6) | 72.0 (69.9–74.2) |
| ImgAlign | 72.3 (69.6–74.8) | 73.6 (70.6–76.5) | 72.3 (69.8–74.5) | 69.7 (66.2–73.4) | 70.2 (68.1–72.1) | 71.2 (68.9–73.4) | 71.7 (69.9–73.4) | 73.9 (71.7–76.0) |
| **ImgFeatAlign** | **74.7 (72.3–77.0)** | **75.0 (72.1–77.7)** | **75.3 (73.1–77.4)** | **72.5 (68.9–75.7)** | **70.4 (68.2–72.3)** | **72.0 (69.6–74.2)** | **72.6 (70.8–74.5)** | **75.2 (73.1–77.5)** |

**Explicit Alignment:** This approach extends the baseline by integrating spatial alignment through deformation fields, improving temporal feature fusion. Following [7], four feature representations are used: current, prior, aligned prior, and their temporal difference, capturing longitudinal changes. Risk is predicted from three inputs—current, prior, and a fused representation that concatenates the current, aligned prior, and difference features. We investigate alignment strategies at both the image and feature levels:

**Feature-Level Alignment (FeatAlign / FeatAlignReg)** learns a deformation field to align prior feature maps, $\mathbf{f}^{\text{pri}}$, to current feature maps, $\mathbf{f}^{\text{cur}}$. FeatAlignReg introduces smoothness regularization to ensure anatomically plausible deformation fields.

**Image-Level Alignment (ImgAlign):** As an alternative to feature-level alignment, we propose MammoRegNet, a deep learning-based registration network inspired by the NICE-Trans architecture [12]. MammoRegNet is used to align prior mammograms, to the current ones. In this setup, current, prior, and aligned prior images are encoded to extract features, from which temporal difference features $\mathbf{f}^{\text{diff}}$ are computed. These features are then passed to the risk prediction module.

**Image-Based Feature Alignment (ImgFeatAlign):** Rather than applying MammoRegNet's deformation field at the image level, this variant applies it directly in feature space. This setup allows us to explore whether image-driven deformation fields can still improve temporal feature fusion when used post-encoding, potentially benefiting from both anatomically grounded registration and deeper feature representations.

## 3 Experimental Setup

**Datasets:** Experiments are conducted on two large public mammography datasets, EMBED [13] and CSAW-CC [14]. Following [7], we include patients with at least 5 years of follow-up. Images are resized to $1664 \times 2048$ (preserving aspect ratio) and split into training, validation, and test sets (5:2:3).

**Evaluation metrics:** Alignment quality is measured by the percentage of Negative Jacobian Determinants [15]. Risk prediction performance is reported using C-index and AUC at 1–5 years [4, 7, 9], with 95% confidence intervals from 1,000 bootstraps. **Implementation Details:** The pre-trained Mirai encoder [9] is used as a frozen backbone. Feature-level alignment jointly optimizes prediction and feature-matching losses, while image-level alignment freezes MammoRegNet. Training uses Adam [16] (LR $1 \times 10^{-5}$, weight decay $1 \times 10^{-6}$, batch size 20) for up to 40 epochs with LR scheduling, early stopping, and augmentations.

## 4 Results

Table 1 summarizes 1- to 5-year breast cancer risk prediction performance (C-index and AUC with 95% CI) for each alignment strategy. ImgFeatAlign consistently achieves the highest C-index and stable AUC, demonstrating superior predictive strength and robustness over time. FeatAlign performs reasonably well but is consistently outperformed by image-level alignment. The Implicit method shows moderate results, while NoAlign yields the lowest scores, with the steepest AUC decline, underscoring the importance of alignment in longitudinal models. These findings highlight the value of advanced alignment strategies for improving the accuracy and reliability of breast cancer risk prediction.

## 5 Conclusion and Outlook

In summary, accurate spatial alignment is crucial for longitudinal breast cancer risk prediction. Image-based approaches, especially ImgFeatAlign, achieve superior performance by balancing anatomical precision with high-level feature representation. These findings highlight the potential of robust longitudinal modeling to enhance personalized screening and early intervention. Future work will extend this by integrating multimodal data to enhance interpretability and risk stratification.

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
