# OpenReview forum: "Reconsidering Spatial Alignment for Longitudinal Breast Cancer Risk Prediction"
_NLDL.org/2026/Abstracts_Track — NLDL 2026 Abstracts_

### Official Review · Reviewer_ekZr · 2025-10-26

**Soundness:** 3
**Correctness:** 3
**Rating:** 4
**Confidence:** 4

**Summary:**

This paper investigates the impact of spatial alignment methods on longitudinal breast cancer risk prediction using mammography. Their findings indicate that image-based feature alignment (ImgFeatAlign) achieves the best performance, balancing anatomical fidelity and predictive robustness. They conclude that accurate spatial alignment substantially improves longitudinal risk prediction, and propose future work integrating multimodal data for interpretability and risk stratification.

**Strengths:**

1. The paper systematically benchmarks multiple alignment strategies (explicit and implicit), providing a fair and structured comparison rarely seen in this domain.
2. The study convincingly argues why alignment matters for longitudinal imaging and supports it with quantitative and anatomical evidence.
3. The proposed ImgFeatAlign model creatively combines anatomical registration and deep feature alignment, leading to state-of-the-art performance.

**Weaknesses:**

1. The study is missing a computational analysis discussing runtime and scalability.
2. The study is missing statistical testing and clinical impact discussion.
3. There is no analysis or visualization of what features or regions the alignment improves.

These are all recommendations for the extended version of this work later :)

Good luck!

---

### Official Review · Reviewer_iM3w · 2025-11-02

**Soundness:** 4
**Correctness:** 4
**Rating:** 5
**Confidence:** 3

**Summary:**

The authors investigate how spatial alignment affects deep learning-based longitudinal breast cancer risk prediction. Using two large mammography datasets, the authors compare explicit (image- and feature-level) and implicit alignment strategies within a unified framework. They propose a novel image-based network, MammoRegNet, that achieves superior performance.

**Strengths:**

The authors evaluate multiple alignment strategies (implicit, feature-level, image-level) within a unified framework, providing clear comparative insights. They use two large, and publicly available datasets (EMBED and CSAW-CC), which strengthens the generalizability of the results. Consistent improvement in both C-index and AUC across multiple time horizons supports the validity and impact of the proposed method.

**Weaknesses:**

While alignment improves prediction performance, the abstract could briefly mention potential clinical implications, such as how improved alignment might integrate into radiology workflows or affect screening intervals. An interesting baseline would be the human scores and relevant discussion on how much time and accuracy difference is between the author’s method and a human judgment.

---

### Decision · Program_Chairs · 2025-11-05

**Decision:**

Accept

**Comment:**

The abstract is of interest to the community and should be presented at the conference.